# Globally consistent assessment of coastal eutrophication

Elígio de Raús Maúre [1✉], Genki Terauchi[1], Joji Ishizaka [2], Nicholas Clinton[3] & Michael DeWitt[3]

Eutrophication is an emerging global issue associated with increasing anthropogenic nutrient loading. The impacts and extent of eutrophication are often limited to regions with dedicated monitoring programmes. Here we introduce the first global and Google Earth Engine-based interactive assessment tool of coastal eutrophication potential (CEP). The tool evaluates trends in satellite-derived chlorophyll-*a* (CHL) to devise a global map of CEP. Our analyses suggest that, globally, coastal waters (depth ≤200 m) covering ~1.15 million km$^2$ are eutrophic potential. Also, waters associated with CHL increasing trends—eutrophication potential—are twofold higher than those showing signs of recovery. The tool effectively identified areas of known eutrophication with severe symptoms, like dead zones, as well as those with limited to no information of the eutrophication. Our tool introduces the prospect for a consistent global assessment of eutrophication trends with major implications for monitoring Sustainable Development Goals (SDGs) and the application of Earth Observations in support of SDGs.

[1] Department of Research and Study, Northwest Pacific Region Environmental Cooperation Center, Toyama, Japan. [2] Institute for Space-Earth Environmental Research, Nagoya University, Nagoya, Japan. [3] Google LLC, 1600 Amphitheater Parkway, Mountain View, CA, USA. ✉email: eligiomaure@gmail.com

Coastal ecosystems provide innumerable services that make them major environmental and economic assets globally[1]. However, their integrity is increasingly threatened by impacts of human activities[2]. Nutrient enrichment, for instance, is known to stimulate phytoplankton productivity. While this growth of phytoplankton can initially be beneficial to the ecosystem, continuous accumulation of organic matter can lead to eutrophication of the system with a series of undesirable ecological effects that can also be harmful to humans. Defined as the increase in the rate of organic matter supply to water bodies[3] the definition of eutrophication has been expanded to meet both scientific and legal requirements[4]. This cultural eutrophication associated with excessive or disproportionate nutrient loading is known for its modifications of nutrient levels and structures including a selective magnification of nitrogen and phosphorus supply but a reduction of silica[5]. These conditions can trigger a chain of biogeochemical feedback including shifts in phytoplankton composition, formation and persistence of harmful algal blooms (HABs) and the consequent occurrence of hypoxic waters[6–9]. The increased incidence of oxygen deficit waters (hypoxic waters), in turn, can stimulate the proliferation of hypoxia-tolerant species such as *Noctiluca scintillans*[10,11]. Furthermore, eutrophication can also increase the possibility of jellyfish outbreaks[12], can contribute to ocean acidification[7] and to degradation of shallow water habitats[13,14]. In the case of submerged vegetation, they can display an array of direct and indirect responses to nutrient loadings that ultimately may lead to their loss, as seen in some seagrass meadows[13]. Therefore, monitoring and/or assessment of eutrophication is important in providing the extent and context of eutrophication[15–17]. Such information is especially relevant for coastal managers to take the required management interventions.

Many coastal regions worldwide experience some level of eutrophication despite that only a few regions with dedicated monitoring programmes have information of eutrophication status. Existing tools for eutrophication assessment[15,18–20], although vital for the identification of eutrophication patterns as well as for understanding the eutrophication causes and consequences, their application entails prohibitively expensive and intensive field monitoring programmes. Alternatively, water quality parameters from satellite imagery are often introduced as effective tools for a synoptic eutrophication assessment[21] and to overcome the spatiotemporal limitations of in situ observations. Chlorophyll-*a* (CHL, mg m$^{-3}$) concentration, a proxy for phytoplankton biomass, is a commonly used indicator of eutrophication as it links nutrient enrichment and the stimulated phytoplankton productivity[22–25]. CHL is recognised by the Global Climate Observing System as an Essential Climate Variable[26,27]. It is an important parameter in the study of the climate system and associated changes, as well as in the study of different factors affecting the dynamics of marine ecosystems including those of anthropogenic origin. In fact, CHL is listed as one of the parameters for the index of coastal eutrophication potential in the Global Manual on Measuring Sustainable Development Goals (SDGs) 14.1.1, 14.2.1 and 14.5.1[28]. To assess coastal eutrophication trends globally, both levels and trends of satellite derived CHL are essential. While the few existing assessment approaches based on satellite data are generally based solely on CHL levels[21,29], the Northwest Pacific Action Plan Eutrophication Assessment Tool (NEAT) considers both the levels and trends of satellite-derived CHL[17,30]. The NEAT was developed by the Special Monitoring and Coastal Environment Assessment Regional Activity Centre (CEARAC) of the Northwest Pacific Action Plan (NOWPAP), a part of the Regional Seas Programme of the United Nations Environment Programme, for the preliminary eutrophication assessment based solely on satellite-derived CHL. It is effective in discriminating both eutrophication potential (see Methods for definitions of eutrophic and eutrophication) waters as well as those in recovery[17].

Our study, therefore, introduces the NEAT as an app constructed on Google Earth Engine (GEE) cloud environment[31] for the global screening of coastal eutrophication potential (CEP). To the best of our knowledge, our app (the Global Eutrophication Watch) is the first of its kind to provide coastal eutrophication trends globally. It classifies CEP based on temporal and spatial patterns of CHL levels as well as trends in annual bloom magnitude allowing for a globally consistent assessment in a way never done before. Although it neither differentiates the bloom forming algae nor determines the frequency or duration of the bloom, it does, however, provide a synoptic view of eutrophic potential waters (those with high levels of CHL) or waters under high risk of eutrophication (those with increasing CHL trends) for prioritised management interventions. The findings not only are pertinent for management and mitigations of eutrophication, but also for monitoring SDGs, specifically indicator 14.1.1a "Index of coastal eutrophication of the SDG 14: Life Below Water". This is to conserve and sustainably use the oceans, seas and marine resources. Further, in addition to putting in-situ obtained results into a wider context, the findings of this study can be put into practice by contrasting them with those obtained from in-situ measurements, model simulations, etc. On the other hand, this study introduces the first global map of CEP for many regions lacking routine water quality monitoring. Accordingly, the information obtained will be vital in guiding the development of monitoring programmes regionally. This study contributes towards the use of Earth Observations in support of the SDGs and the results emphasize the importance of the Global Eutrophication Watch as a global framework for eutrophication monitoring.

## Results and discussion

We first introduce a case study in Bohai Sea (Fig. 1a)—a semi-enclosed marginal sea, one of the China seas, that has been severely impacted by human activities in the last half century—to demonstrate the value of the eutrophication screening tool (cf. 2.1). The Bohai Sea has become eutrophic and suffers from symptoms of eutrophication that are well-documented[8,9,32]. Second, we introduce the global screening of CEP in section 2.2 using the satellite data from the Moderate Resolution Imaging Spectroradiometer on Aqua (MODISA), reprocessing 2018, with a spatial resolution of 4 km, obtained using the standard ocean colour index algorithm (OCI[33]; https://oceancolor.gsfc.nasa.gov/atbd/chlor_a/). The CHL time series from MODISA are the longest among ocean colour sensors and are used as the default data for the global assessment.

The OCI algorithm provides adequate CHL retrieval in the global open ocean. In optically complex coastal waters, like in the Bohai Sea, however, the optically active constituents (e.g., coloured dissolved organic matter) and phytoplankton may vary independently[34], so reliable CHL retrievals may not be achieved[35,36]. Therefore, in 2.1 we adopted a CHL product that uses a regional algorithm. This regional product was obtained using the Yellow Sea Large Marine Ecosystem Ocean Color Project (YOC) algorithm, an empirical algorithm appropriate for the Bohai Sea as it alleviates the impacts of suspended sediments and coloured dissolved organic matter on CHL retrievals[35]. The YOC CHL data span a 22-year period (1998-2019) and have a spatial resolution of 1 km. Further details of the datasets are given in Methods, 3.1.

The definitions adopted for the terms eutrophic and eutrophication potential, as discussed in the following sections, are

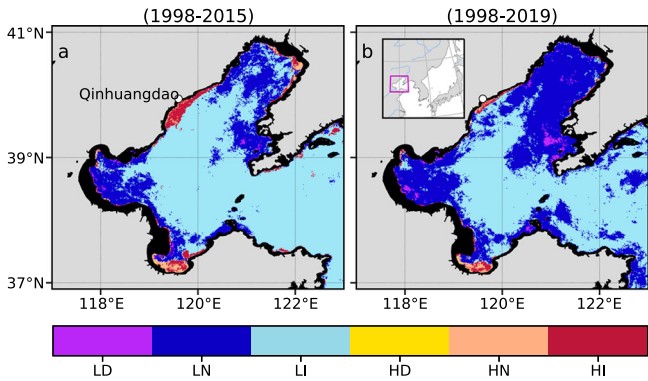

**Fig. 1 Map of coastal eutrophication potential (CEP) in the Bohai Sea.** LD, LN, and LI depict the status as being low CHL ($\alpha < 5$ mg m$^{-3}$) with decreasing trend, no trend and increasing trend, respectively. HD, HN and HI indicate high CHL ($\alpha \geq 5$ mg m$^{-3}$) with the three above-mentioned trends, respectively. **a** Preliminary assessment of CEP for the period 1998–2015. **b** Same as **a** but for the period 1998–2019. The rectangle in magenta (**b**) shows the location of Bohai Sea.

given in the Methods section (3.3). Briefly, eutrophic potential will refer to any productive system characterised by high CHL, whereas eutrophication potential refers to the process of becoming eutrophic or a progression of an already eutrophic water body. Oligotrophication potential will be the reverse of eutrophication potential.

**Assessment of coastal eutrophication potential: a case study in the Bohai Sea.** The assessment results obtained from the global eutrophication watch (Fig. 1) comparing two assessment periods, 1998–2015 and 1998-2019, revealed that some coastal waters associated with high CHL ($\geq 5$ mg m$^{-3}$) and increasing temporal trends (HI) have significantly shrunk. In contrast, low CHL ($<5$ mg m$^{-3}$) waters with no trends associated (LN) and low CHL waters with decreasing trends (LD) have expanded in the Bohai Sea. Overall, the area covered by pixels associated with increasing trends, that is, eutrophication potential (LI and HI) shrank ~27%, whereas for those associated with oligotrophication potential (LD and HD) had a threefold increase between the two assessment periods (Fig. 1). This, in part, may be indicative of improving water quality. Reports suggest that there have been a series of control measures implemented in China to reduce nutrient emissions from terrestrial sources. These measures curbed the worsening trend of coastal eutrophication in the China seas[25], and possibly contributed to the gradual decrease in red tides annual frequency and dramatic decrease in the red tides affected area since 2003[9,25]. Besides the human interventions discussed above, natural climate variability also plays a role in the variations of CHL[37]. Zhai et al.[37] discussed the influence of sea surface temperature and rainfall on CHL long-term changes. Warmer temperature anomalies, present during the positive phase of the Pacific Decadal Oscillation (PDO), and concurrent with negative rainfall anomalies in the Bohai Sea, were suggested to be conducive to negative CHL anomalies through their negative impacts on nutrient fluxes into the sea. Transient factors such as water exchange between the Bohai Sea and the Yellow Sea also modulate the variations of CHL[38]. Additionally, the observed changes in CHL trends between the two assessment periods could also be an indication of the sensitivity of trend detection to the data length. This study highlights the usefulness of our tool, which identifies the spatial patterns of eutrophication potential using CHL levels and trends over a larger spatial and temporal scales and condensed in a single map.

The patterns identified in Fig. 1 are further corroborated by reports of water quality of the Bohai Sea. As already stated, most of the bays in this sea have been severely impacted by human activities. Particularly, high inputs of dissolved inorganic nitrogen have been observed in recent decades[5,8,9]. Interestingly, the increase in nitrogen inputs continued even when the total river discharge consistently decreased in relation to that of the 1960s[9]. As a result, several ecological disasters such as the incidence of red-tides and the occurrence of hypoxia and/or anoxia intensified[9,39]. In the case of the assessment map in Fig. 1, a patch of HI was effectively identified in the coastal waters off Qinhuangdao. This region has been found to be an oxygen minimum zone in the Bohai Sea. Further, the waters adjacent to this patch constitute a hypoxia hotspot[8]. The above illustrates the suitability of our tool in identifying the spatial distribution of CEP with the application of CHL from satellite ocean colour remote sensing.

The number of pixels associated with CHL increasing trends significantly decreased between the two assessment periods (Fig. 1) indicating that there might be some large-scale phenomena driving this shrinkage of increasing trends in the whole Bohai Sea. Atmospheric deposition, which acts on a much larger scale, can be an important nutrient source to the ocean[40]. In the Bohai Sea, the influence of atmospheric deposition is also significant. Observations and simulation results suggest that the atmospheric contribution to dissolved inorganic nitrogen can range from ~25% to 54% of the total[39,41]. As for the flux of particulate phosphorus entering the sea through windblown dust storms, it can be >500 times greater than on normal days[39]. In recent years, however, declines in the frequencies of dust storms and the volume of China's emissions of major anthropogenic air pollutants have been observed[39,42]. The decline in emissions results from the introduction of China's clean air policies in 2010, driving significant reductions in pollutant emissions in the first seven years of its inception[42]. As shown in a modelling study assessing the effects of atmospheric nitrogen deposition on the marine ecosystem in the Bohai Sea[41], the inclusion of the atmospheric deposition can cause an average increase in phytoplankton biomass of >50%. It naturally follows that our latter assessment (Fig. 1b), which includes recent years when both dust storms and anthropogenic emissions have markedly reduced, might reflect the long-term changes in atmospheric nutrient deposition.

Other large-scale climate processes such as El Niño (La Niña) and PDO have also been implicated in the dynamics of the Bohai Sea ecosystem[37,38]. Fan et al.[38] analysed the spatial and temporal variations of particulate organic carbon (POC) in the Yellow-Bohai Sea over the period 2002–2016. They suggested that the above climate indices impact the surface POC through their influence on water exchange between the Yellow-Bohai Sea and the East China Sea. This water exchange is controlled by the East Asian winter monsoon and its influence on the Yellow Sea Warm Current. The fact that these factors appear to have an indirect influence[38] suggest that atmospheric deposition might be a major driver of the observed large-scale decrease in CHL levels and trends.

As introduced above, the study by Zhai et al.[37] used a 16-year record of MODISA CHL and observed spatially coherent increasing CHL trends from 2003 to 2011 and decreasing trends from 2012 to 2018 in the Bohai Sea. They suggested that these changes were mainly controlled by variations in sea surface temperature and rainfall, which are linked to the PDO. In positive PDO phases, positive temperature (negative rainfall) anomalies prevail in the Bohai Sea. These conditions lead to decreased dissolved inorganic nitrogen content in the surface layers due to suppressed vertical nutrient diffusion and reduced land-sourced

nutrient fluxes[37]. Factors such as changes in nutrient levels and structures have major impacts on CHL long-term changes. Wang et al.[39] showed that the summer concentration of dissolved inorganic nitrogen in the Bohai Sea continuously increased from the 1990s, while that of phosphorus exhibited a decreasing trend in the period 1978–2016. So, the nitrogen/phosphorus ratio mostly followed that of nitrogen content[39]. As a result, the nutrient regime of the Bohai Sea has shifted from nitrogen-limitation before the 1990s to potential phosphorus-limitation thereafter[8,9,39].

Although we speculate about the possible factors driving the CHL variability observed in the Bohai Sea, the changes in nutrients levels and structures as well as the CHL response in a eutrophic environment are complex[5,9]. At this point, we emphasize that our procedure is simply meant for the screening of CEP. The mechanisms behind the identified patterns are beyond the scope of the tool and that should be supplemented by follow-up studies. Here, we stress the use of CHL estimates from ocean colour remote sensing as the preliminary parameter for a rapid and a consistent assessment of CEP globally. The significance of this approach is in the use of a single parameter that condenses the spatial and temporal information which allows the identification of areas in potential need of preventive management or eutrophication mitigation efforts.

**Assessment of coastal eutrophication potential: global ocean.** The global map of CEP (Fig. 2a) is composed mostly of LN and HI (Table 1). Pixels associated with high CHL are mostly found in coastal and inland waters. Here, we only focus on the coastal waters (depth ≤200 m). To get an intuition of the global distribution of area covered by each eutrophication potential waters, the area estimate was obtained through the combined use of bathymetry data and the marine biogeochemical provinces[43]. Our analysis suggested that globally LI and HI (~799,305 km$^2$) occupy a larger fraction of coastal waters than LD and HD (~602,406 km$^2$). The major fraction of both LD-HD and LI-HI combined was found in coastal provinces of Asia (SUND, Table 1). However, the HI class was predominant in the Atlantic Ocean where some of the well-known dead zones, the Gulf of Mexico and the Baltic Sea, are found[16,44]. Besides the above cases, there are many other coastal seas which were flagged as eutrophication potential (both LI and HI) and are distributed across the globe (Table 1). Although Table 1 also includes coastal upwelling regions, their contribution is relatively smaller than non-upwelling regions. These examples emphasize the utility of the introduced tool in preliminary eutrophication assessment. Not only was the tool able to identify known areas of eutrophication, but also those potentially suffering from the effects of eutrophication in addition to non-reported locations experiencing some level of eutrophication[6,45]. Therefore, the introduction of our Global Eutrophication Watch, a rapid and consistent preliminary assessment of CEP is now globally feasible. This tool should instigate a concerted action against the proliferation of coastal eutrophication.

In addition to the global map of eutrophication potential, we also compared the assessment results based on our improved CHL introduced in 2.1 vs. the standard MODISA CHL product for the period 2003–2019. Overall, we found the CEP waters identified with YOC CHL (Fig. 2b) were also apparent in the standard MODISA product (Fig. 2c, d). However, LD waters appeared more than LI in the map generated using the standard CHL. The retrievals of CHL in highly dynamic and optically complex waters such as in coastal waters are challenging. The existing algorithms for atmospheric correction are robust in the open ocean where the ocean colour covaries with phytoplankton concentration[46]. In the case of Bohai Sea, we have the YOC and

some other statistically based CHL retrieval algorithms[36] that best represent the phytoplankton variability. We believe that different regions may also have a CHL product that more accurately suits the characteristics of the designated area. The global application of our methodology in preliminary assessment of CEP should not be contingent on the global standard CHL product. In our GEE-based tool, the Global Eutrophication Watch, there is an option for users to enter the path to their asset (dataset in the GEE) of monthly CHL time series. This monthly CHL data can then be used in the assessment instead of the default datasets.

While the focus is on the preliminary assessment of eutrophication potential, oligotrophication potential (LD, HD) are equally worthy of mention. Under the warming climate, the tropics and subtropics are likely to experience enhanced stratification and reduced nutrient supply to the euphotic layer. As a result, phytoplankton growth will be limited with long-term decline (Fig. 2a) associated with decreasing primary production[47]. In coastal and enclosed seas, measures to reduce nutrient loading can lead to decreased phytoplankton concentration or reduce the eutrophication and associated ecological disasters such as the incidence of hypoxic events, though other issues like oligotrophication can emerge[48]. The Seto Inland Sea of Japan experienced severe eutrophication during the high economic growth period of the 1960s and 1970s[49], but now is reported to be undergoing oligotrophication[48]. Significant reductions in nutrient loading along with loss in biodiversity are reported to be the precursors of oligotrophication. Moreover, in the oligotrophication process, changes in the food web structure are suggested to have caused a decrease in fishery production of the Seto Inland Sea[50].

In this study, we introduced the Global Eutrophication Watch, a tool for a preliminary eutrophication assessment solely based on satellite-derived CHL. Although different eutrophication assessment methods exist, especially comprehensive eutrophication assessment methods[15,19,20], their global application is complicated by the need for extensive and intensive field observation campaigns. So, the significance of our introduced tool is in its simplicity and scale. It only uses satellite derived CHL to provide a systematic assessment of CEP at a macroscopic (global) and microscopic (regional) levels as well as with sufficient temporal information to allow coastal water managers make informed decisions on where to focus their eutrophication management efforts. In this method, we stress the importance of CHL levels and trends. This combination provides a simple but robust assessment scheme. For instance, low CHL but increasing trends (LI) may inform managers about required management actions to prevent future ecological disasters. This warning might go missing in case only CHL levels[21] are considered. On the other hand, with the sole use of CHL trends, high CHL but no trend waters (HN) can be overlooked. CHL levels are often linked to phytoplankton biomass, which is also linked to the health of the ecosystem. So, our methodology is inexpensive and robust for a global assessment of CEP.

Overall, we expect this contribution to aid in the many global efforts acting to counter the impacts of nutrient pollution and eutrophication. It is well known that management planning efforts should also incorporate available knowledge, and adapt to changing environmental conditions, while evaluating the effectiveness of implemented measures. Thus, our Global Eutrophication Watch tool, with its ready-to-use map of up-to-date information of the status of CEP, provides the required scientific knowledge to support monitoring programmes, adaptive management, and decision-making. It is also useful for educational purposes and in raising awareness, as it is simple and uses very few resources. A simple internet connection, either on a smartphone or computer, allows one to evaluate eutrophication trends worldwide.

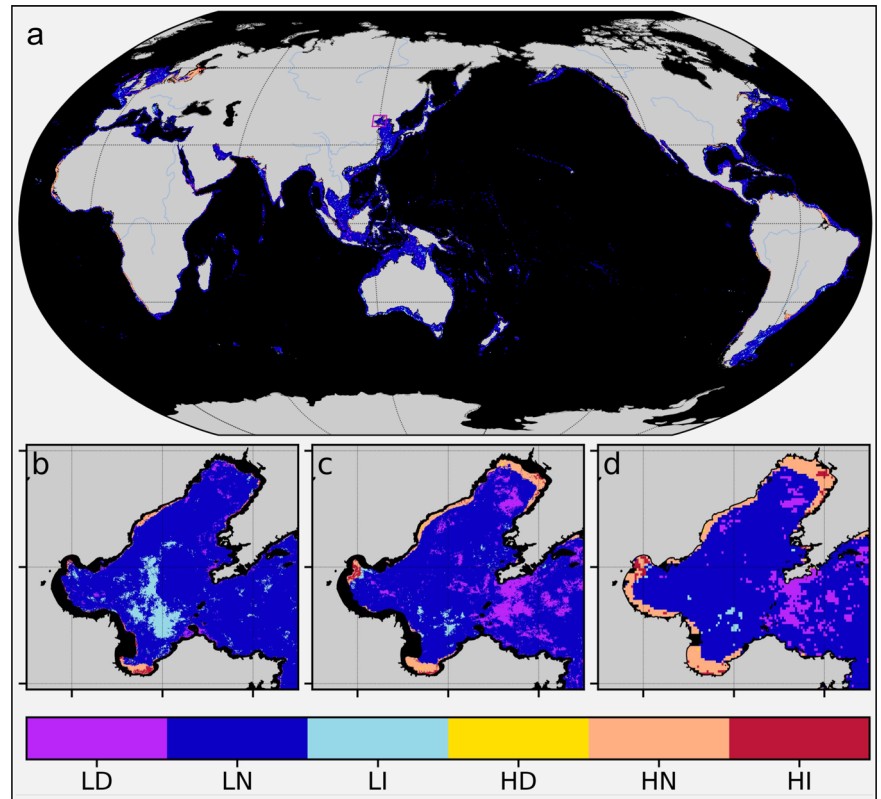

**Fig. 2 Map of CEP in the global ocean and in the Bohai Sea. a** Preliminary assessment of CEP in the global ocean for the period 2003–2019 based on MODISA global dataset. The CHL threshold is same as in Fig. 1. **b** Preliminary assessment of CEP in the Bohai Sea based on the YOC algorithm for the same period as in **a** but with spatial resolution of 1 km. **c** Same as **b** but for MODISA 1 km spatial resolution. **d** Same as **b** but for MODISA 4 km spatial resolution. The southern and northern regions with few observations (<70% in the 17-year period) were masked. The GEE App is accessible through the link https://eutrophicationwatch.users.earthengine.app/view/global-eutrophication-watch.

**Table 1 Global distribution of the area coverage estimates of each eutrophication potential class based on the coastal biogeochemical provinces (https://www.marineregions.org/sources.php#longhurst) as in Longhurst (2006)[43].**

**Total area estimate [km²]**

| Province name | Code | Ocean | LD | LN | LI | HD | HN | HI |
|---|---|---|---|---|---|---|---|---|
| Alaska Downwelling Coastal Province | ALSK | Pacific | 9920 | 183,293 | 2827 | 148 | 8787 | 227 |
| Australia-Indonesia Coastal Province | AUSW | Indian | 9652 | 806,935 | 69,319 | 202 | 3490 | 327 |
| Benguela Current Coastal Province | BENG | Atlantic | 437 | 43,847 | 7938 | 1355 | 67,213 | 4542 |
| Brazil Current Coastal Province | BRAZ | Atlantic | 6424 | 416,745 | 43,420 | 659 | 57,375 | 10,164 |
| California Upwelling Coastal Province | CCAL | Pacific | 15,017 | 64,733 | 1328 | 903 | 30,438 | 1137 |
| Canary Coastal Province | CNRY | Atlantic | 4481 | 121,800 | 5994 | 3323 | 74,124 | 2981 |
| Caribbean Province | CARB | Atlantic | 23,965 | 711,105 | 41,763 | 1464 | 75,830 | 15,257 |
| Central American Coastal Province | CAMR | Pacific | 51,493 | 168,289 | 6862 | 496 | 5196 | 103 |
| Chile-Peru Current Coastal Province | CHIL | Pacific | 6559 | 108,325 | 3615 | 1450 | 46,428 | 2435 |
| China Sea Coastal Province | CHIN | Pacific | 38,415 | 863,543 | 56,532 | 595 | 31,508 | 4546 |
| East Africa Coastal Province | EAFR | Indian | 9790 | 257,185 | 6347 | 148 | 7435 | 262 |
| East Australian Coastal Province | AUSE | Pacific | 6048 | 289,005 | 18,183 |  | 58 |  |
| East India Coastal Province | INDE | Indian | 15,941 | 214,052 | 4760 | 738 | 20,183 | 248 |
| Guianas Coastal Province | GUIA | Atlantic | 12,525 | 354,734 | 22,682 | 2800 | 122,381 | 9079 |
| Guinea Current Coastal Province | GUIN | Atlantic | 4119 | 150,981 | 25,115 | 1109 | 85,136 | 11,468 |
| Kuroshio Current Province | KURO | Pacific | 7552 | 326,373 | 25,766 | 371 | 4696 | 89 |
| Mediterranean Sea, Black Sea Province | MEDI | Atlantic | 10,327 | 426,123 | 38,027 | 481 | 5458 | 448 |
| New Zealand Coastal Province | NEWZ | Pacific | 4312 | 85,754 | 2859 |  |  |  |
| Northeast Atlantic Shelves Province | NECS | Atlantic | 45,583 | 696,349 | 36,761 | 3568 | 190,774 | 19,073 |
| Northwest Arabian Upwelling Province | ARAB | Indian | 14,759 | 128,285 | 4123 | 2,117 | 30,154 | 1188 |
| Northwest Atlantic Shelves Province | NWCS | Atlantic | 27,986 | 834,160 | 39,927 | 1082 | 55,701 | 2847 |
| Red Sea, Persian Gulf Province | REDS | Indian | 56,697 | 353,303 | 4,188 | 417 | 4,681 | 477 |
| Southwest Atlantic Shelves Province | FKLD | Atlantic | 11,093 | 709,303 | 94,027 | 13 | 698 | 141 |
| Sunda-Arafura Shelves Province | SUND | Pacific | 166,659 | 3,298,508 | 125,220 | 2,758 | 71,585 | 7181 |
| West India Coastal Province | INDW | Indian | 13,337 | 248,278 | 16,975 | 3120 | 28,645 | 528 |

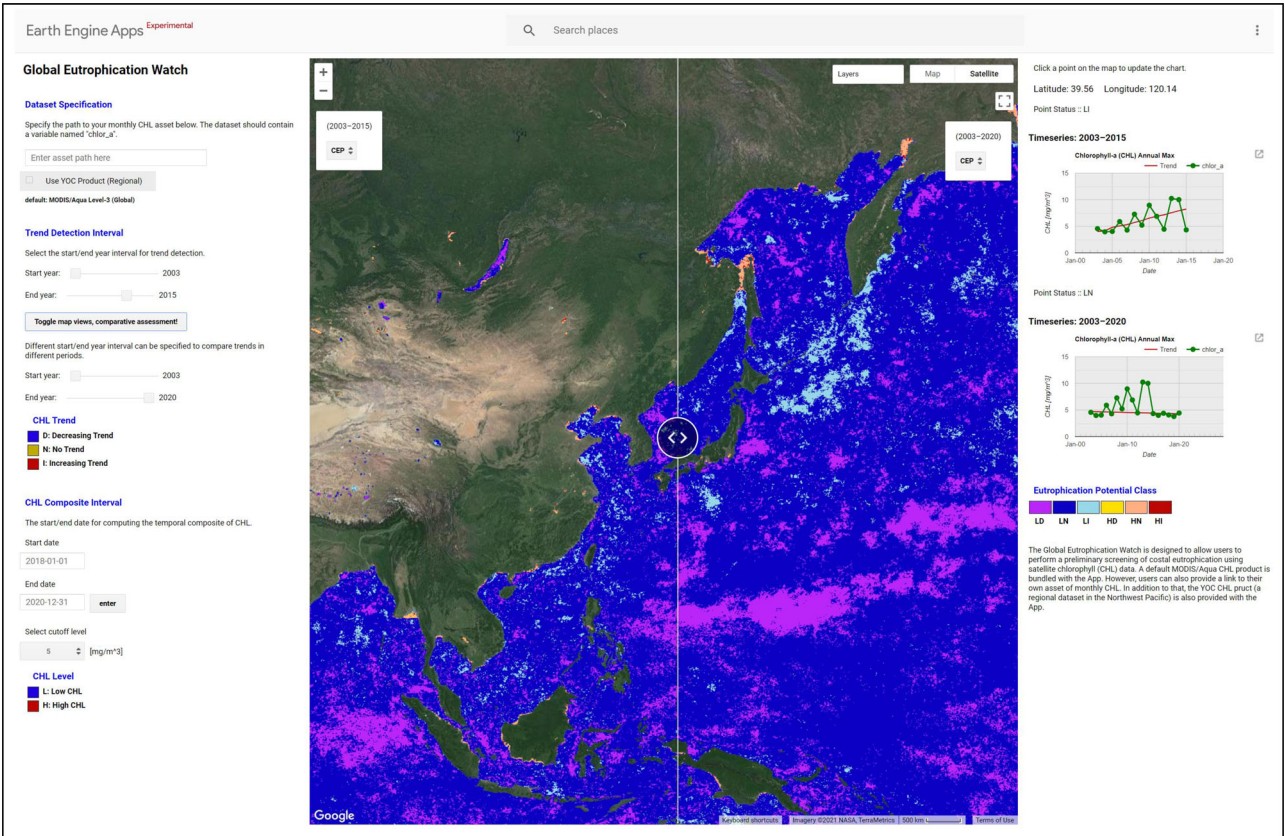

**Fig. 3 Global eutrophication watch app interface.** The left panel shows the control panel of the app. The map of CEP in the NOWPAP region based on MODISA global dataset is shown in the middle. The right panel shows the CHL max time series of a select point on the map. The CEP classes are also shown.

## Methods

**CHL datasets.** For the global detection of CEP, we used the currently available 17-year record of daily CHL data from the Moderate Resolution Imaging Spectro-radiometer on Aqua (MODISA), reprocessing 2018 (https://oceancolor.gsfc.nasa.gov/reprocessing/r2018/aqua/), and with a spatial resolution of 4 km. The data set is stored in the App's asset (see 3.4 below) and its temporal extension is updated on a yearly basis. In addition to these yearly updates, the data sets will also be updated following NASA (National Aeronautics and Space Administration, U.S.) periodic reprocessings that improve product quality with advances in algorithms or sensor calibration knowledge.

Besides the global screening of CEP, a case study was developed in the Bohai Sea (cf. 2.1) to demonstrate the usefulness of the introduced tool. In coastal regions, like the Bohai Sea, CHL estimates based on the standard ocean colour algorithms—like the MODISA OCI algorithm (https://oceancolor.gsfc.nasa.gov/atbd/chlor_a/) —often fail. The reason being that the optical properties of these complex waters (also denoted case 2 waters) are influenced not just by phytoplankton but other optically active constituents (including coloured dissolved organic matter), and these may vary independently of one another[34]. So, for this case study, we used a regionally tuned CHL dataset obtained as monthly composites from the Marine Environmental Watch of the NOWPAP (https://ocean.nowpap3.go.jp/) with spatial resolution of 1 km. These CHL data were only available in the NOWPAP region.

The above regional CHL data were obtained using the YOC algorithm, designed to alleviate the impacts of coloured dissolved organic matter and suspended sediments on CHL retrievals[35]. The YOC product, as used in this study, was a blending of YOC CHL and CHL based on OC algorithm combined with the colour index (CI) algorithm, that is, the OCI algorithm[33]. The switching between the two was determined by the values of normalised water leaving radiance (nLw, mWcm$^{-2}$ mm$^{-1}$ sr$^{-1}$) at 555 nm[51]. The YOC algorithm was applied in waters with high nLw555 (>2.5), whereas the OCIs were applied in waters with low nLw555 (<1.5). A smooth transition between the two extremes was ensured by a linear combination in the mid-range of nLw555 (2.5>nLw555 > 1.5). Accordingly, adequate CHL estimates could be obtained in waters with high nLw that otherwise would be overestimated[35,51], and in such cases the YOC CHL had superior quality with better spatial and temporal variations relative to the standard products[51–53]. Therefore, this improved CHL is of critical importance to the case study presented here. The YOC algorithm was originally developed using the Sea-viewing Wide Field-of-view Sensor (SeaWiFS) sensor bands[35]. Its application to MODISA and to Medium Resolution Imaging Spectrometer (MERIS) data was based

on the regression between SeaWiFS and MODISA (MERIS) bands and band ratios. In this study we used the CHL data based on the OCI algorithm. Please refer to Terauchi et al.[17] for additional details on the computation of regression coefficients.

Given that the global level 3 data constitute our default asset for the global eutrophication assessment, following the case study in the Bohai Sea (2.1), in 2.2 we briefly compare the trends estimated using the YOC CHL with those obtained from the global data readily obtainable from NASA. This comparison is essential given that the NASA global standard products are more accessible than any other lower-level (such as level 2) data to non-expert users, including water quality managers and decision makers. In addition, it is the least expensive way for a rapid eutrophication assessment before a thorough, in-situ based investigation can follow.

**Trend analysis.** The estimation of trends at pixel level is based on the Sen's slope method[54]—a non-parametric trend estimation method—which detects the presence of monotonic trends in a yearly data record at the 90% significance level. Nonparametric tests provide higher statistical power in the case of nonnormality, as is the case with CHL, and are robust against outliers and large data gaps. Trends estimated below a critical threshold are treated as N (no trend). Moreover, as the focus is on the detection of eutrophication potential with consideration of it being a process occurring over a long-time scale (on the order of years), the temporal trends in CHL are estimated from annual maximum from monthly composites of each considered year. The choice is partly motivated by the fact that the evaluation of existence of monotonic trends can also be statistically challenged by short-term variability in CHL. So, by using CHL annual maximum from monthly composites, we effectively remove the seasonal and short-term variabilities. Doing so, we focus on the CHL peak season. Consequently, the obtained trends reflect the interannual behaviour of the phytoplankton bloom season, assuming that the bloom is manifested as high biomass.

**NEAT methodology as a global screening tool of coastal eutrophication.** In this study we used the NEAT methodology to develop a GEE-based tool for the global detection of CEP (the Global Eutrophication Watch) using satellite-derived CHL. In its screening procedure, the NEAT—a robust satellite-based preliminary assessment tool of eutrophication potential—unifies, in a single map, the temporal and spatial information of the area under consideration. It combines the levels and trends of CHL to generate six patterns of water quality[17]. The CHL levels generate two

patterns based on the CHL concentration ($\alpha$ [mg m$^{-3}$]), the first being composed by CHL lower than the threshold $\alpha$, CHL < $\alpha$ (L), and the other by CHL ≥ $\alpha$ (H). The trends have three patterns, namely: waters with decreasing trend (D), with no trend (N), or with increasing (I) trend. In this way, a composite map of six classes can be generated, viz. LD, LN, LI, and HD, HN, HI. Before moving on to the explanation of the meaning of each class, it is worth defining the terms adopted in this paper for clarity. Eutrophic potential will be used to indicate a productive system with high CHL, whereas eutrophication potential refers to the process of becoming eutrophic or a progression of an already eutrophic water body. In addition to the above definitions, we also introduce oligotrophication potential which is associated with the progression to a least productive water body. Hence, pixels flagged HD, HN, HI are eutrophic potential with HD indicative of systems under recovery, whereas in LI and HI are eutrophication potential. In HI, the conditions may worsen as the water body is already eutrophic potential. Moreover, LD is suggestive of reversed eutrophication, that is, further oligotrophication. LN and HN are indicative of L and H CHL but stable conditions over the analysis period.

It is important to note that classification of waters as being L or H is subject to the consideration of the threshold $\alpha$, which will vary depending on the conditions of each region. However, the same is not the case for D, N or I. Trends will most probably be impacted by the length of the analysis period and/or other environmental factors controlling the variability of CHL rather than a given $\alpha$. As such, both LD, LI and HD, HI provide critical information about the eutrophication of the system under scrutiny. The global eutrophication watch, therefore, not only provides important information of areas potentially in need of preventive management efforts, but also helps in evaluating the impacts of measures taken to reduce the effects of eutrophication. The NEAT procedure uses a threshold of 5 mg m$^{-3}$, and this threshold is computed based on the most recent 3-year mean data of the analysis period. Nevertheless, this threshold is not fixed, and users are able to adjust the level and the composite period to area specific values as different regions may have different thresholds according to the region's background.

In the above-introduced approach we model the interannual changes in phytoplankton bloom magnitude (CHL annual maximum) to assess the eutrophication of a coastal ecosystem. The main purpose is the identification of waters with symptoms of coastal eutrophication, which may include the incidence of HABs or other related issues[5,14,25]. Although HABs can be a direct or indirect manifestation of eutrophication, the interactions between the two are complex[5,9,24]. HABs, often, associate with specific types of algal blooms such as cyanobacteria, *Karenia* spp., etc[29,55,56]. But CHL, which we used for our index of eutrophication, is present in both HAB and non-HAB blooms. Although satellite-derived CHL has been found, in some cases, sufficient to detect HABs[29,56], often additional information (such as the knowledge of local CHL patterns) is necessary to make the link between the two. So, while our approach can identify patterns of algal bloom magnitude over the years and relate them to eutrophication potential, neither does it discriminate the bloom forming algae nor does it determine the frequency or duration of the bloom. As such, this approach can only provide a context of areas with symptoms of water quality deterioration, and therefore with potential for HAB occurrence (cf. 2.1) without a priori information of the considered ecosystem.

Successful HAB detection or prediction often goes beyond the sheer use of CHL, and in most cases in-situ observations or more complex approaches are required. For example, Stumpf et al.[29] used CHL anomaly to flag waters with potential for *Karenia brevis* blooms in the Gulf of Mexico. Their CHL anomaly was computed as a difference between a single image of satellite-derived CHL and a two-month average image taken two weeks prior to the image being considered. While their procedure was successful in *Karenia brevis* identification in the Gulf of Mexico, such methodology would be limited in other environments with different background or with a different *Karenia* species[29,55].

The merits of our eutrophication screening approach are in its use of CHL levels and trends. If only the CHL trends are considered, the magnitude of the problem would be overlooked. On the other hand, if only the levels are considered, only the spatial dimension of the problem would be captured[21,29,57] and thereby overlooking, for instance, LI waters. So, here we reemphasise the importance of the spatial and temporal dimensions provided by satellite derived CHL and condensed in a single map by this approach, which retains both space-time information. Thus, a synoptic view of eutrophication potential is gained prior to any expensive field sampling, although vital to complementing satellite information.

**The GEE Global Eutrophication Watch App.** The Global Eutrophication Watch (Fig. 3) on the GEE is composed of three main fields: (1) the data-set specification panel, (2) the panel for selection of trend detection intervals and (3) the specification of the CHL composite interval and the threshold selection panels. The data panel allows the selection of two default data sets, that is, MODISA and YOC CHL. In practice, only YOC CHL can be checked as MODISA is the *de facto* default. Moreover, this panel also includes a box for users to enter the path to an Earth Engine asset of monthly composites of CHL for the tool to read and use for the assessment. The option is especially important given the challenges associated with CHL retrievals in the coastal waters. Unlike in the open ocean, where phytoplankton dominate the optical properties or co-vary with other optically active constituents, in coastal waters phytoplankton may vary independently of the optical constituents, and thus the global CHL product may fail to resolve

phytoplankton variations[58]. So, this option can be understood as a plug-in that allows users around the globe to conduct the eutrophication assessment based on their own datasets. This feature enables users to incorporate regionally improved CHL data while keeping the assessment procedure consistent. This has the immediate result of allowing consistent results to be obtained from a spectrum of ecosystems with different characteristics. The next panel is used to specify the trend detection interval, the start and end years. This panel also includes a button to toggle views, that is, to split the map into two windows providing a capability for comparative assessment. The impact of inclusion of more years in the trend detection analysis, for instance, can be verified by simply using two different year intervals. Finally, the last user defined parameters are for the CHL threshold. Controls for start and end dates are available for users to indicate the time interval to be used to compute the mean CHL. This is used in conjunction with the cut-off level (threshold) to split L vs. H CHL waters.

## Data availability
The satellite derived CHL data used in (2.2) are available from the NOWPAP Marine Environmental Watch website at https://ocean.nowpap3.go.jp/. The data used in (2.3) are available from the website of the NASA's Ocean Biology Processing Group at https://oceancolor.gsfc.nasa.gov/. The eutrophication potential maps in Fig. 1 through 3 can be obtained via the Google Earth Engine global eutrophication watch app at https://eutrophicationwatch.users.earthengine.app/view/global-eutrophication-watch The data of coastal biogeochemical provinces are available from the Marine Regions at https://www.marineregions.org/sources.php#longhurst. The bathymetry map used in combination with biogeochemical provinces was created using Windows Image Manager (https://www.wimsoft.com/).

## Code availability
The Earth Engine code used for trend analysis based on the Sen's slope method is available at the Google Earth Engine community tutorials (https://developers.google.com/earth-engine/tutorials/community/nonparametric-trends).

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

## Acknowledgements

This work was made possible through the support received from the Ministry of the Environment of Japan, the Toyama Prefectural Government, and the Northwest Pacific Region Environmental Cooperation Center established to promote the Action Plan for the Protection, Management and Development of the Marine and Coastal Environment of the Northwest Pacific Region as a part of the Regional Seas Programme of the United Nations Environment Programme. E.R.M. thanks Tak Amaru for improving the language quality of the introduction.

## Author contributions

E.R.M. and G.T. conceived the study. G.T. and J.I. developed the remote sensing-based eutrophication assessment method. E.R.M. processed and ingested the satellite derived CHL datasets into Earth Engine. N.C. implemented the Sen's slope detection method on Earth Engine. E.R.M. led the Earth Engine App development with support from M.D. E.R.M led the writing of the manuscript with contributions from all authors.

## Competing interests

The authors declare no competing interests.
