## [Peer Review File · Nature Communications]

Reviewers' comments:

Reviewer #1 (Remarks to the Author):

Review of "Satellite Chlorophyll and Google Earth Engine based Global Eutrophication Inspector"

Summary:

This paper describes an index of ocean "eutrophication" based on satellite chlorophyll records in google earth engine (GEE), and the change in that index with time. The relevance and limitation of this eutrophication inspector is discussed. The inspector is provided as an App in GEE.

Comments:

Satellite ocean color measurements are highly sensitive to instrument and vicarious calibration changes. NASA's ocean color data record is periodically reprocessed as new knowledge is gained on the calibration, and as algorithms are updated to improve performance. These updates can significantly effect biases and trends. Currently, the MODIS/Aqua record is at version R2018.0. There is no indication as to the processing version used for the analysis in this paper, nor is there any version details on the GEI App or the datasets described in Google Earth Engine. In fact, it is not clear if a consistent dataset is being used, or if a mix of versions is included. It is unclear if or when the retrospective data in Google Earth Engine is updated following a reprocessing by NASA. These issues are not entirely the fault of the authors, but using and interpreting data from GEE without ensuring version consistency has the potential to be highly misleading. Please: 1) verify that the data record in use is a consistent version (perhaps get the data directly from NASA as a verification), 2) discuss how the GEE/GEI would be updated when the next NASA reprocessing occurs, 3) discuss the potential impact of instrument calibration on the derived GEI trends.

It seems as if this GEI App was developed on the basis of ready-availability of global, long-term NASA chlorophyll datasets in GEE. The datasets available are low-resolution, mapped equirectangular projection (not equal area). As the authors admit, they are not ideal for studying coastal phenomena such as eutrophication, where higher-resolution, as is available from Level-2 products distributed directly by NASA, would provide more detailed information with a well known provenance. Perhaps this paper should focus on the description of the App and its value and limitations, rather than the interpretation of results which may be more accurately derived from other available datasets, or alternatively it should use the best available information for the analysis and discuss the App as an interesting application with limitations.

The authors fold low chlorophyll regions (LI) into the statistics and state that "this indicates that potential eutrophication or areas with potential to become eutrophic are globally pervasive with no signs of recovery.

Thus, appropriate measures or policies are necessary to counter the effects of excess nutrient loading that deteriorate the ecosystems.” Is this LI specific to the coastal/inland-water regions (<200m)? As introduced at the start of this paragraph, LI covers much of the global deep oceans, where Chl is very low and where increased productivity is often considered a good thing (e.g., to support the ecosystems and sequester global carbon). Please clarify that all discussion starting on Line 76 is for shelf and inland waters. If it is not, then this discussion should be deleted.

The results for High Chl regions seem to suggest that the primary interpretation is that the trends are neutral. Why do the authors repeatedly interpret HN as areas “with potential to become eutrophic”?

Regarding the trend analysis, there is no information here on trend uncertainties based on uncertainties in the data source. How confident are these HI and HD trends? Are LI and LD trends equally confident? Does the period over which the trends were derived greatly impact the results (e.g., including or not the last year or two, where calibrations trends are less certain), impact of major climate-driven oscillations occurring in the last year. Please discuss the trend uncertainties in greater detail.

Chl threshold of 5 is a very blunt hammer. Would the conclusions change significantly if the threshold was 3 or 7? Please discuss the sensitivity of results to the chosen threshold.

There are many typos and grammatical issues (missing plurals, missing articles, missing commas): e.g.,

L70, immediate of preventive

L122, of potential eutrophic were identified

L125, largest on planet and

L132, 25 percent of the global.

L146, is capture as HD

L156, that contribute large fraction of

L178, the GEI uses trend of annual CHL max we

L179, One of the reason for

L180, based on locally tuned algorithm

L183, such as near river mouth where

L238, suited to study the coastal ecosystems that dynamic and can be optically complex

L264, waters in near-future.

L273, construct a tool usefulness for preliminary assessment

Reviewer #2 (Remarks to the Author):

The paper presents an interesting and innovative approach to the creating a global context of coastal eutrophication bases on massive data available from near continuous satellite remote sensing data. There are obviously issues related to the accuracy and comparability of such data that specialists would criticize and the authors discuss some of them. The assumptions regarding the categorization of potentially eutrophic or not by an arbitrary cut-off of 5 mg m⁻³ can also be criticized, but the authors also indicate that this could be adjusted based on another criterion. As a demonstration of the approach, this seems reasonable. I have a few suggestions regarding how the manuscript could be improved in addition to by-line editorial comments:

Clarify the scope of the coverage and be consistent throughout. At various places this is stated as coastal waters, coastal and inland waters, and waterbodies. While the Caspian Sea is discussed and is an inland waterbody, does the global analysis also include large freshwater bodies such as the North American and African great lakes? Also, it should be recognized that there is much attention around the world on eutrophication of bays and estuaries, such small coastal water bodies less than a certain scale cannot be assessed by this approach.

Explain in more detail why the maximum of monthly composites is a reasonable basis for characterization. Are there ways that the user could elect other criteria?

More clearly exclude naturally occurring high chlorophyll areas from the analysis and statements concerning potential eutrophic areas and trends. There are areas clearly not enriched from anthropogenic sources and for

which we would not expect improving or declining trends. Be careful in labeling HN areas as likely to decline with or without actions.

Acknowledge that trends over the period 2003 to 2018 could reflect the influence of climactic cycles such as ENSO, PDO, and NAO, as well as eutrophic trends or secular climate change.

7 Eutrophication of what? Coastal waters? Should more clearly define the scope in the title, abstract and introduction.

15-16 It doesn't logically follow that the problem (eutrophication) is likely persist or worsen in the absence of countermeasures in the areas that have no trend associated. In those areas, why is not also as likely that it will improve as worsen?

28-30 Could and probably should add loss of submerged vegetation to this list.

46-48 There is something wrong with this sentence. "concentration of chl-a . . . provides estimates of satellite chl-a"?

49 NOWPAP undefined, Northwest Pacific Action Plan?

53 Information on the extent of eutrophication?

77 Does this just refer to just the coastal (defined by <200 m dept, as per Methods) and inland waters or all HD, HN, HI waters?

84-86 This sentence needs to be rethought and clarified to be consistent with the previous one. If HD is included how can one state they have no signs of recovery? Also, these estimates include areas of high natural CHL biomass, such as upwelling zones where the notion of eutrophication and recovery do not apply.

86-87 This statement should be qualified in light of my previous comment.

100 This is a rather dated reference and not from the scientific literature.

113 One should not leave the impression that west Africa is the only place where CHL concentrations are naturally high because of upwelling. The eastern margins of the Pacific basin are also noteworthy upwelling zones.

161 Maybe just conditions are not improving rather than unlikely to improve.

170-173 Consider the recent paper by Wang et al. (2019. Science: 365:83) that suggests Amazon nutrients may be stimulating Sargassum growth offshore.

236 data WERE used.

238 that ARE dynamic.

240 data ARE useful.

249 (recent and future) more progress in . . . IS still needed . . .

253 have been proposed, both . . .

270 level IS largely . . . or LEVELS are largely . . .

275 globally, BUT future . . .

277 current data PROVIDE . . .

Reviewers' comments:

Reviewer #1 (Remarks to the Author):

Review of "Satellite Chlorophyll and Google Earth Engine based Global Eutrophication Inspector"

Summary:

This paper describes an index of ocean "eutrophication" based on satellite chlorophyll records in google earth engine (GEE), and the change in that index with time. The relevance and limitation of this eutrophication inspector is discussed. The inspector is provided as an App in GEE.

Comments:

Satellite ocean color measurements are highly sensitive to instrument and vicarious calibration changes. NASA's ocean color data record is periodically reprocessed as new knowledge is gained on the calibration, and as algorithms are updated to improve performance. These updates can significantly effect biases and trends. Currently, the MODIS/Aqua record is at version R2018.0. There is no indication as to the processing version used for the analysis in this paper, nor is there any version details on the GEI App or the datasets described in Google Earth Engine. In fact, it is not clear if a consistent dataset is being used, or if a mix of versions is included. It is unclear if or when the retrospective data in Google Earth Engine is updated following a reprocessing by NASA. These issues are not entirely the fault of the authors, but using and interpreting data from GEE without ensuring version consistency has the potential to be highly misleading. Please: 1) verify that the data record in use is a consistent version (perhaps get the data directly from NASA as a verification), 2) discuss how the GEE/GEI would be updated when the next NASA reprocessing occurs, 3) discuss the potential impact of instrument calibration on the derived GEI trends.

1) We appreciate the reviewer for bringing this point into our attention. We have verified the dataset on GEE and unfortunately found that it is in fact a mix of different versions. However, the maps that were submitted with the paper were obtained from the data that we directly downloaded from NASA and they were version R2018.0. As for the data currently used by the revised App, we have uploaded R2018 into the App's asset. More discussion on this point is included in the materials and methods.

2) Please see L295 of this revised manuscript where we mentioned about possible updates following NASA reprocessing.

3) The trends obtained by our App can be significantly impacted by uncalibrated data as the uncalibrated data can include degradation and aging artifacts, for instance, that are known to

introduce significant bias on timeseries analysis as in the case of trend estimations. NASA shows that instrument calibration contributes to a significant bias shift of order 2-10% on the global deep-water (<https://oceancolor.gsfc.nasa.gov/reprocessing/r2018/aqua/>). So, in our Earth Engine App, we use the reprocessing R2018, which is the recommended and incorporates the most recent data quality improvements performed by NASA. Please refer to L293-298 where we mention about the processing version and the data provenance.

In the previous submission, we devoted significant effort in discussing issues associated with data quality as we were primarily using the global level-3 MODIS-Aqua data as the main product for our assessment. The reason was that our primary focus is coastal eutrophication, but the data we were using was the global level-3 data that performs poorly in the coastal regions. Some of these limitations are still discussed in the material and methods section (L320).

So, instead of extending this discussion about the data quality, which we believe shifts the attention of our paper, in the revised manuscript, we use the global level-3 data solely for demonstrations purposes. As such, we have worked a way to give users the possibility of incorporating their best quality data for the eutrophication assessment. We believe this is the best approach given that different coastal regions have different characteristics and the issues impacting the quality of data vary by location. Please see L322-326.

It seems as if this GEI App was developed on the basis of ready-availability of global, long-term NASA chlorophyll datasets in GEE. The datasets available are low-resolution, mapped equiarectangular projection (not equal area). As the authors admit, they are not ideal for studying coastal phenomena such as eutrophication, where higher-resolution, as is available from Level-2 products distributed directly by NASA, would provide more detailed information with a well known provenance. Perhaps this paper should focus on the description of the App and its value and limitations, rather than the interpretation of results which may be more accurately derived from other available datasets, or alternatively it should use the best available information for the analysis and discuss the App as an interesting application with limitations.

Thank you for the comments. In this revision of the manuscript, we removed most of the interpretation of results obtained from the use of global level-3 data. Instead, we introduce a case study in the Bohai Sea. We use this case study to show the value of our procedure and App as a tool for preliminary eutrophication assessment globally. Please refer to 2.1 (L96) and L299-311.

Moreover, in this case study, we use a chlorophyll product from a locally tuned algorithm (the YOC algorithm) that improves the quality of the data over the default NASA standard product. The YOC algorithm is applied directly to level-2 products directly obtained from NASA.

The authors fold low chlorophyll regions (LI) into the statistics and state that “this indicates that potential eutrophication or areas with potential to become eutrophic are globally pervasive with no signs of recovery. Thus, appropriate measures or policies are necessary to counter the effects of excess nutrient loading that deteriorate the ecosystems.” Is this LI specific to the coastal/inland-water regions (<200m)? As introduced at the start of this paragraph, LI covers much of the global deep oceans, where Chl is very low and where increased productivity is often considered a good thing (e.g., to support the ecosystems and sequester global carbon). Please clarify that all discussion starting on Line 76 is for shelf and inland waters. If it is not, then this discussion should be deleted.

Thank for the comment. To avoid further confusion, in the revised manuscript, the open ocean has been masked and only coastal waters are shown, that is, depth ≤ 200 m. This is also stressed in L166.

The results for High Chl regions seem to suggest that the primary interpretation is that the trends are neutral. Why do the authors repeatedly interpret HN as areas “with potential to become eutrophic”?

Thank for the comment. In fact, HN should not be interpreted as areas with potential to become eutrophic. For clarity, the HN in the revised manuscript is used to indicate eutrophic potential waters as opposed to eutrophication potential. Please see L257-259 for the definitions adopted in the manuscript concerning the use of eutrophic versus eutrophication.

Regarding the trend analysis, there is no information here on trend uncertainties based on uncertainties in the data source. How confident are these HI and HD trends? Are LI and LD trends equally confident? Does the period over which the trends were derived greatly impact the results (e.g., including or not the last year or two, where calibrations trends are less certain), impact of major climate-driven oscillations occurring in the last year. Please discuss the trend uncertainties in greater detail.

Regarding the trend analysis, we use the Sen’s method for the trend estimation (L278). In the paper and in the App, we consider a 90% significance level. So, trends estimated below this critical threshold are treated as no trend.

Discussion about the inclusion of the last two or more years in the trend analysis is covered in the case study we introduce in 2.1 where we compare the results obtained over two assessment periods 1998-2015 and 1998-2019. Please see L96.

Chl threshold of 5 is a very blunt hammer. Would the conclusions change significantly if the threshold was 3 or 7? Please discuss the sensitivity of results to the chosen threshold.

Changing the threshold (which we call cut-off level in the App) only shifts waters between the two categories characterising the levels of chlorophyll concentration, that is, low or high chlorophyll concentration waters. However, the total number of pixels initially classified as eutrophication potential (LI and HI) remain the same. The only thing happening is a shift from either LI to HI or vice-versa. So, based on the definition of eutrophication (L258), these changes have no impact on the conclusions. In addition, we have kept the threshold in the App adjustable from the start because we recognise that no single value can fold different environmental conditions into a single group. Please see some discussion on the issue in L265-270.

There are many typos and grammatical issues (missing plurals, missing articles, missing commas): e.g.,

We have carefully checked for these typos and grammatical issues.

The following comments have all been taken into consideration. The manuscript has undergone extensive revision so as to eliminate many typos and grammatical issues. Most of the sentences indicated in the following lines have been revised or completely modified/deleted.

L70, immediate of preventive

L122, of potential eutrophic were identified

L125, largest on planet and

L132, 25 percent of the global.

L146, is capture as HD

L156, that contribute large fraction of

L178, the GEI uses trend of annual CHL max we

L179, One of the reason for

L180, based on locally tuned algorithm

L183, such as near river mouth where

L238, suited to study the coastal ecosystems that dynamic and can be optically complex

L264, waters in near-future.

L273, construct a tool usefulness for preliminary assessment

Reviewer #2 (Remarks to the Author):

The paper presents an interesting and innovative approach to the creating a global context of coastal eutrophication bases on massive data available from near continuous satellite remote sensing data. There are obviously issues related to the accuracy and comparability of such data that specialists would criticize and the authors discuss some of them. The assumptions regarding the categorization of potentially eutrophic or not by an arbitrary cut-off of 5 mg m⁻³ can also be criticized, but the authors also indicate that this could be adjusted based on another criterion. As a demonstration of the approach, this seems reasonable. I have a few suggestions regarding how the manuscript could be improved in addition to by-line editorial comments:

Thanks for your positive comments, we appreciate it.

Clarify the scope of the coverage and be consistent throughout. At various places this is stated as coastal waters, coastal and inland waters, and waterbodies. While the Caspian Sea is discussed and is an inland waterbody, does the global analysis also include large freshwater bodies such as the North American and African great lakes? Also, it should be recognized that there is much attention around the world on eutrophication of bays and estuaries, such small coastal water bodies less than a certain scale cannot be assessed by this approach.

Explain in more detail why the maximum of monthly composites is a reasonable basis for characterization. Are there ways that the user could elect other criteria?

A discussion on this issue has been added in L282-291.

Currently, this in the default way of doing the assessment and this is basically the procedure we are proposing as the methodology for preliminary eutrophication assessment.

More clearly exclude naturally occurring high chlorophyll areas from the analysis and statements concerning potential eutrophic areas and trends. There are areas clearly not enriched from anthropogenic sources and for which we would not expect improving or declining trends. Be careful in labeling HN areas as likely to decline with or without actions.

Thank you for the comment. The discussion of the global distribution of coastal eutrophication potential was revised altogether. In the revised manuscript, we only provide a table with area coverage estimates in different coastal biogeochemical provinces. Please see L184.

Acknowledge that trends over the period 2003 to 2018 could reflect the influence of climactic cycles such as ENSO, PDO, and NAO, as well as eutrophic trends or secular climate change.

Thank you. We have provided a more focused discussion in the case study we introduce in 2.1 in which we discuss a few possible factors controlling the long-term variability of chlorophyll in the Bohai Sea. Please see L96.

7 Eutrophication of what? Coastal waters? Should more clearly define the scope in the title, abstract and introduction.

Thank you. We have made this point clear throughout the text.

15-16 It doesn't logically follow that the problem (eutrophication) is likely persist or worsen in the absence of countermeasures in the areas that have no trend associated. In those areas, why is not also as likely that it will improve as worsen?

We have revised the use of the eutrophication term in the manuscript to remove this ambiguity. In the revised manuscript, we use the term eutrophication potential (L258) to more concisely indicate waters classified and LI, HI.

28-30 Could and probably should add loss of submerged vegetation to this list.

Added. Please see L47.

46-48 There is something wrong with this sentence. "concentration of chl-a . . . provides estimates of satellite chl-a"?

The sentence has been revised entirely.

49 NOWPAP undefined, Northwest Pacific Action Plan?

Thank you. NOWPAP is now defined in L66.

53 Information on the extent of eutrophication?

The sentence has been removed.

77 Does this just refer to just the coastal (defined by <200 m dept, as per Methods) and inland waters or all HD, HN, HI waters?

Originally HD, HN, HI waters only referred to those within the depth <200 m depth, as per Methods. In the revised manuscript, this area estimate for the eutrophication potential waters was revised to only include LI, HI waters within the depth ≤ 200 m.

84-86 This sentence needs to be rethought and clarified to be consistent with the previous one. If HD is included how can one state they have no signs of recovery? Also, these estimates include areas of high natural CHL biomass, such as upwelling zones where the notion of eutrophication and recovery do not apply.

The text was revised, and this point has been made clear. Eutrophication potential waters in this revision do not include HD. HD waters are considered eutrophic potential as opposed to LI, HI waters which are classified as eutrophication potential. Please see L257-259 for the definitions adopted in the manuscript concerning the use of eutrophic versus eutrophication.

86-87 This statement should be qualified in light of my previous comment.

Revised. Thank you.

100 This is a rather dated reference and not from the scientific literature.

Revised. Thank you.

113 One should not leave the impression that west Africa is the only place where CHL concentrations are naturally high because of upwelling. The eastern margins of the Pacific basin are also noteworthy upwelling zones.

Thank you for the comment. This discussion has been removed in the revised manuscript.

161 Maybe just conditions are not improving rather than unlikely to improve.

Revised. Thank you.

170-173 Consider the recent paper by Wang et al. (2019. Science: 365:83) that suggests Amazon nutrients may be stimulating Sargassum growth offshore.

Thank you for the suggestion. In the revised manuscript, we no longer discuss details of the global distribution of coastal eutrophication. Instead, we introduce a case study in Bohai Sea with more focused discussion on eutrophication. Please see 2.1, L96.

Thank you for the comment. All the following issues have been taken into consideration.

236 data WERE used.

238 that ARE dynamic.

240 data ARE useful.

249 (recent and future) more progress in . . . IS still needed . . .

253 have been proposed, both . . .

270 level IS largely . . . or LEVELS are largely . . .

275 globally, BUT future . . .

277 current data PROVIDE . . .

REVIEWER COMMENTS

Reviewer #2 (Remarks to the Author):

This is a substantially very different manuscript than the one I reviewed in November 2019. The method remains generally similar, although there are refinements, in part in response to the comments of reviewers of the earlier manuscript. In general, the authors were adequately responsive to the criticisms and suggestions of this reviewer. However, in addition to the global assessment of the earlier manuscript, the authors have added a substantial case study on the Bohai Sea as a test of concept. This addition adds considerable value to the manuscript, but it made reconsideration of the manuscript more difficult than just reviewing the revision to determine its responsiveness to the reviews. My following comments deal specifically with the new Bohai Sea results and their interpretation.

L97-100. Is the shrinkage in the HI areas along the Qinhuangdao area just because CHL values fell below 5 mg/m³ from 2015 to 2019? This could give the impression that the changes were more dramatic than they were and give a false impression that CHL was now decreasing (from red to green).

L 103-106. The authors should at least briefly mention here other possible, transient factors that would affect changes in CHL and trends during the 2015–2019 period, such as exchange with the Yellow Sea (Fan et al.) that is discussed later on. Reference 21 (Wang et al. 2018) summarized the development of coastal eutrophication in the China seas, did not conclude that it had been reduced. Specifically, they stated that “the worsening trend has been curbed but the status of coastal eutrophication has not been substantially improved”.

L131–147. The focus on changes in the atmospheric deposition of both N and P as a cause of the changes in CHL is intriguing, but it should be noted that there are no direct measurements of changes in deposition on the Bohai Sea or commensurate declines in surface water concentrations. There are other major changes that could affect CHL concentrations that should also be mentioned, particularly the decline in delivery of P from the rivers even as N discharges continued to rise. Several authors have pointed to changes in N/P ratios in surface waters to conclude that the system shifted to P-limitation.

Beyond what might occur in the editorial process leading to publication, there are a number of problems with the references that require the authors' attention: Reference 11 (Goes et al 2018) is incomplete and requires the title and page numbers for this chapter in the referenced book. References 15, 17 are incomplete and require information on how to access these documents (URL or doi). Reference 38 requires the title of the book in which this chapter appeared. In reference 40 there is no need for “in” as this is a journal article.

Reviewer #3 (Remarks to the Author):

Review of “Globally consistent assessment of coastal eutrophication” by E.R. Maúre, G. Terauchi, J. Ishizaka, N. Clinton, and M. DeWitt

The Authors present an index of aquatic eutrophication based on satellite ocean color chlorophyll data records incorporated into a Google Earth Engine app. Overall, I find the work to be compelling and the GEE app to be very cool. I have two lingering issues whose resolution I believe will support an improved manuscript prior to publication.

First, the flow of the presentation is somewhat cumbersome and lacking, at least for a reader who wants to know more about the datasets considered prior to seeing the results. I realize the materials and methods should appear last in this format, but the satellite considered isn't even mentioned until after the first suite

of results is presented. Furthermore, many useful satellite details are missing. My recommendation is that (1) the authors briefly introduce the datasets in the introduction and (2) they take more care to present details on the satellite datasets in the materials and methods (e.g., temporal resolution isn't mentioned and a figure caption is currently the only place where consideration of different spatial footprints is considered).

Second, the choice of statistical analysis is discussed, but not fully justified or placed in context of what it reveals and what it doesn't address. I'm not suggesting changes, but feel the authors should be more descriptive and conscientious in the discussion of their statistical metrics (see specific comments below).

Regarding previous reviewer comments and author responses: I believe the revision captures and largely addresses the comments made by the first series of reviewers. That said, I think some of my comments build upon previous reviewers' concerns.

Ultimately, I'm in favor of this work and support its publication after further revision.

Specific comments:

Line 54: Suggest listing units to variables when they are first introduced (e.g., mg/m³).

Line 76: Suggest rewording to "using satellite-derived CHL data".

An issue I have with the way this manuscript flows is that the satellite dataset isn't mentioned until after the results are presented. I'm not suggesting an exhaustive description early on (that belongs where it is in Section 3), but it's confusing to me to not even know that this is MODISA data being used – sometimes with the standard algorithm, sometimes with a specialized algorithm. Line 186, e.g., is the first time MODISA is mentioned, and paragraphs of results are already presented. This is also the first time that the alternate algorithm is mentioned. I realize that YOC is described later, but still think you need to make some clear introductory mention earlier on of what datasets are being presented in the results.

Line 186: Need to be more clear that this is a specialized algorithm with improved results in certain water types. Not a generically "improved" algorithm. Generally speaking, the utility of specialized/regional vs. global algorithms needs to be introduced earlier than the materials and methods section. Otherwise, the impact and interpretation of results is unclear.

Line 196: Reword to "standard CHL product. In our ..."

Line 204: This is the only time variations in spatial resolution are mentioned. Need to explain this further, at least in the materials and methods. See later questions about YOC.

Line 265: To shorten this long paragraph and create a multiple sentence paragraph for the following one, suggest a paragraph break at "It is important ..." and combine with the paragraph that begins on line 274.

Lines 286-291: I understand the authors' rationale for evaluating the annual max CHL from monthly composites. However, I think their discussion is incomplete. While their metric evaluates changes in bloom abundance, it doesn't touch on bloom frequency or duration. While these may not be significant parameters for open ocean blooms, they are critical metrics of HAB occurrences. For completeness, this should be discussed and the justification for using the annual max should be better placed in context of what it describes and what it doesn't. Also, I wonder if the authors considered evaluating trends in annual CHL anomalies? I expect that could also provide useful information. Perhaps worthwhile to justify their method in the context of other potential methods as well.

Line 294: What temporal resolution? Clarify monthly.

Line 301: Reference for the standard algorithm?

Line 302: Are these NOWPAP data also at 4 km? What temporal resolution? Is YOC globally available?

Line 309: Need to state clearly under what conditions the "superiority" is achieved.

RESPONSE TO REVIEWER COMMENTS

Reviewer #2 (Remarks to the Author):

This is a substantially very different manuscript than the one I reviewed in November 2019. The method remains generally similar, although there are refinements, in part in response to the comments of reviewers of the earlier manuscript. In general, the authors were adequately responsive to the criticisms and suggestions of this reviewer. However, in addition to the global assessment of the earlier manuscript, the authors have added a substantial case study on the Bohai Sea as a test of concept. This addition adds considerable value to the manuscript, but it made reconsideration of the manuscript more difficult than just reviewing the revision to determine its responsiveness to the reviews. My following comments deal specifically with the new Bohai Sea results and their interpretation.

Thank you. We appreciate the positive feedback.

In the following lines we report the revisions made in the text. The lines referenced to the manuscript with track change.

L97-100. Is the shrinkage in the HI areas along the Qinhuangdao area just because CHL values fell below 5 mg/m³ from 2015 to 2019? This could give the impression that the changes were more dramatic than they were and give a false impression that CHL was now decreasing (from red to green).

The reported shrinkage was based on the trends identified between the two assessment periods. The number of pixels with increasing trends is smaller in 1998-2019 period as compared with 1998-2015.

The figure below is taken from our global eutrophication watch app. The time series on the right are from the red dot in the vicinity of Qinhuangdao area. In the figure we compare two periods, 1998-2015 and 1998-2020. It is obvious that there is an inflection point after the max is observed around 2014. Later years after 2014 show a decreasing trend. Of course, the levels have also decreased in the two periods so that the composite of the latter years is in most cases below 5 mg m⁻³. But the point is that not only the red but also the green patches have decreased.

Our observations of decreasing trends are corroborated by a recent study that also found decreasing trends in recent years in the Bohai Sea (Zhai et al. 2021). They used MODIS-Aqua derived CHL and found a spatially coherent upward trend over 2003–2011 and downward trend over 2012–2018.

We have revised the wording to make the point clearer. Please see line 143 (manuscript with track changes).

L 103-106. The authors should at least briefly mention here other possible, transient factors that would affect changes in CHL and trends during the 2015–2019 period, such as exchange with the Yellow Sea (Fan et al.) that is discussed later on. Reference 21 (Wang et al. 2018) summarized the development of coastal eutrophication in the China seas, did not conclude that it had been reduced. Specifically, they stated that “the worsening trend has been curbed but the status of coastal eutrophication has not been substantially improved”.

Thank you for the comment. We have added other possible aspects that would affect changes in CHL. In addition, we have also revised the wording to remove the impression caused by our previous sentence regarding the status of coastal eutrophication trends reported by Wang et al. 2018.

Please see the revisions in lines 148-159.

L131–147. The focus on changes in the atmospheric deposition of both N and P as a cause of the changes in CHL is intriguing, but it should be noted that there are no direct measurements of changes in deposition on the Bohai Sea or commensurate declines in surface water concentrations. There are other major changes that could affect CHL concentrations that should also be mentioned, particularly the decline in delivery of P from the rivers even as N discharges continued to rise. Several authors have pointed to changes in N/P ratios in surface waters to conclude that the system shifted to P-limitation.

The reviewer is correct about the shift of the Bohai Sea ecosystem to P-limitation. We agree that not only changes in the atmospheric deposition, but also other factors play a role in CHL

changes in the Bohai Sea. We have expanded the discussion to consider these additional factors. Please refer to lines 209-221.

Beyond what might occur in the editorial process leading to publication, there are a number of problems with the references that require the authors' attention: Reference 11 (Goes et al 2018) is incomplete and requires the title and page numbers for this chapter in the referenced book. References 15, 17 are incomplete and require information on how to access these documents (URL or doi). Reference 38 requires the title of the book in which this chapter appeared. In reference 40 there is no need for "in" as this is a journal article.

Thank you for the comments. We have revised all citation information.

Reference

Zhai, F. et al. Interannual-decadal variation in satellite-derived surface chlorophyll-a concentration in the Bohai Sea over the past 16 years. *J. Mar. Syst.* 215, 103496 (2021). doi:<https://doi.org/10.1016/j.jmarsys.2020.103496>

Reviewer #3 (Remarks to the Author):

Review of “Globally consistent assessment of coastal eutrophication” by E.R. Maúre, G. Terauchi, J. Ishizaka, N. Clinton, and M. DeWitt

The Authors present an index of aquatic eutrophication based on satellite ocean color chlorophyll data records incorporated into a Google Earth Engine app. Overall, I find the work to be compelling and the GEE app to be very cool. I have two lingering issues whose resolution I believe will support an improved manuscript prior to publication.

We appreciate the positive feedback.

We reply to comments below with indication of revisions (line number) referred to the manuscript with track change.

First, the flow of the presentation is somewhat cumbersome and lacking, at least for a reader who wants to know more about the datasets considered prior to seeing the results. I realize the materials and methods should appear last in this format, but the satellite considered isn't even mentioned until after the first suite of results is presented. Furthermore, many useful satellite details are missing. My recommendation is that (1) the authors briefly introduce the datasets in the introduction and (2) they take more care to present details on the satellite datasets in the materials and methods (e.g., temporal resolution isn't mentioned and a figure caption is currently the only place where consideration of different spatial footprints is considered).

We are sorry for the cumbersomeness in the presentation. We have revised the manuscript to address these issues.

- (1) We briefly introduce the datasets in lines 91-107 (manuscript with track changes).
- (2) A thorough discussion of the datasets is now introduced in materials and methods 3.1 (line 344).

Second, the choice of statistical analysis is discussed, but not fully justified or placed in context of what it reveals and what it doesn't address. I'm not suggesting changes, but feel the authors should be more descriptive and conscientious in the discussion of their statistical metrics (see specific comments below).

Thank you for the comment. We have added the description of what our metric reveals, and the limitations associated in 3.3 (line 449).

Regarding previous reviewer comments and author responses: I believe the revision captures and largely addresses the comments made by the first series of reviewers. That said, I think some of my comments build upon previous reviewers' concerns.

Ultimately, I'm in favor of this work and support its publication after further revision.

Thank you for your positive feedback, we appreciate it.

Specific comments:

Line 54: Suggest listing units to variables when they are first introduced (e.g., mg/m³).

Thank you. Units added.

Line 76: Suggest rewording to “using satellite-derived CHL data”.

Thank you. Suggestion adopted throughout the text.

An issue I have with the way this manuscript flows is that the satellite dataset isn't mentioned until after the results are presented. I'm not suggesting an exhaustive description early on (that belongs where it is in Section 3), but it's confusing to me to not even know that this is MODISA data being used – sometimes with the standard algorithm, sometimes with a specialized algorithm. Line 186, e.g., is the first time MODISA is mentioned, and paragraphs of results are already presented. This is also the first time that the alternate algorithm is mentioned. I realize that YOC is described later, but still think you need to make some clear introductory mention earlier on of what datasets are being presented in the results.

Thank you for the valuable suggestion. We have provided a brief description of the datasets prior to the results section. Please see lines 91-107.

Line 186: Need to be more clear that this is a specialized algorithm with improved results in certain water types. Not a generically “improved” algorithm. Generally speaking, the utility of specialized/regional vs. global algorithms needs to be introduced earlier than the materials and methods section. Otherwise, the impact and interpretation of results is unclear.

Thank you. This comment has also been addressed along with the previous one in lines 91-107. Further discussion was added to materials and methods 3.1 (lines 344).

Line 196: Reword to “standard CHL product. In our ...”

Revised, thank you.

Line 204: This is the only time variations in spatial resolution are mentioned. Need to explain this further, at least in the materials and methods. See later questions about YOC.

The spatial resolution of each dataset is now introduced in lines 91-107. More description has also been added to materials and methods 3.1 (line 344).

Line 265: To shorten this long paragraph and create a multiple sentence paragraph for the following one, suggest a paragraph break at “It is important ...” and combine with the paragraph that begins on line 274.

Revised, thank you.

Lines 286-291: I understand the authors' rationale for evaluating the annual max CHL from monthly composites. However, I think their discussion is incomplete. While their metric evaluates changes in bloom abundance, it doesn't touch on bloom frequency or duration. While these may not be significant parameters for open ocean blooms, they are critical metrics of HAB occurrences. For completeness, this should be discussed and the justification for using the annual max should be better placed in context of what it describes and what it doesn't. Also, I wonder if the authors considered evaluating trends in annual CHL anomalies? I expect that could also provide useful information. Perhaps worthwhile to justify their method in the context of other potential methods as well.

Thank you for the comment.

We have expanded the discussion to address what the metric describes and what it does not. Please refer to discussion in lines starting at 449.

As for the CHL anomalies, we anticipate that the results obtained, either using the absolute values or anomalies, will remain consistent. Unless a more specialised algorithm is used to find a specific pattern, departures from the current results are not expected.

For example, Stumpf et al. (2003) used satellite derived CHL anomaly to flag waters with potential for *karenia brevis* blooms in the Gulf of Mexico. Their CHL anomaly is obtained through a difference between a single image and a two-month average image taken two weeks prior to the image being considered. In such case, they were successful in finding the patterns of anomalous CHL associated with *karenia brevis*.

The following figures show our eutrophication assessment maps obtained using absolute values (top) and annual CHL anomalies (bottom). As stated above, the results remain consistent in the two figures. Departures from these patterns may appear depending on how the anomalies are computed. But such deviations would simply emphasise the difference in approaches taken and most importantly in what kind of information is being isolated.

Here, we would like to emphasize that our aim was to identify symptoms of eutrophication using CHL. These may include HABs, but the study is not particularly focusing on HAB detection of any kind. HAB detection should benefit from more advanced or focused algorithms, and these algorithms are difficult to be applied globally as in the example of Stumpf et al.

Line 294: What temporal resolution? Clarify monthly.

The temporal resolution of each dataset has been added. Please refer to line 348 and line 361. “Monthly” modified to “monthly composites of CHL”.

Line 301: Reference for the standard algorithm?

Reference added in line 93.

Line 302: Are these NOWPAP data also at 4 km? What temporal resolution? Is YOC globally available?

Sorry for the confusion. The NOWPAP data are 1 km and monthly temporal resolution. The YOC is a NOWPAP regional product. We have expanded the description of the datasets to make these points clearer. Please refer to 3.1 (line 345).

Line 309: Need to state clearly under what conditions the “superiority” is achieved.

Thank you for the valuable comment.

The superiority is achieved when the influence of suspended sediment and coloured dissolved organic matter is pronounced. These points have been briefly mentioned in the paragraph 95 and further discussed in materials and methods (please refer to 3.1, paragraph line 345).

Reference

Stumpf, R. P. et al. Monitoring *Karenia brevis* blooms in the Gulf of Mexico using satellite ocean color imagery and other data. *Harmful Algae* 2, 147–160 (2003).
doi:[https://doi.org/10.1016/S1568-9883\(02\)00083-5](https://doi.org/10.1016/S1568-9883(02)00083-5)

REVIEWERS' COMMENTS

Reviewer #2 (Remarks to the Author):

The manuscript adequately addresses the issues I raised in my review of the first revision.

Reviewer #3 (Remarks to the Author):

Re-review of "Globally consistent assessment of coastal eutrophication" by E.R. Maúre, G. Terauchi, J. Ishizaka, N. Clinton, and M. DeWitt

This is a very nice revision of the manuscript and the Authors have satisfactorily addressed all of my comment. Congrats to them on this successful work. I now find this suitable for publication, pending two very minor questions:

1. Did you acquire "chlorophyll concentration" or "chlorophyll concentration – OCx algorithm" from the NASA OBPG? If the latter, you're all set. If the former, this is not exclusively OC3M, but rather the OCI (CI + OC3M blend) from Hu et al. 2012 and 2019. Just thought this should be clarified.

2. Should Karenia be capitalized?

All the best.